# Placental Characteristics of a Large Italian Cohort of SARS-CoV-2-Positive Pregnant Women

**DOI:** 10.3390/microorganisms10071435

**Published:** 2022-07-15

**Authors:** Michele Antonio Salvatore, Edoardo Corsi Decenti, Maria Paola Bonasoni, Giovanni Botta, Francesca Castiglione, Maria D’Armiento, Ezio Fulcheri, Manuela Nebuloni, Serena Donati

**Affiliations:** 1National Centre for Disease Prevention and Health Promotion, Istituto Superiore di Sanità—Italian National Institute of Health, Viale Regina Elena 299, 00161 Rome, Italy; michele.salvatore@iss.it (M.A.S.); serena.donati@iss.it (S.D.); 2Department of Biomedicine and Prevention, University of Rome Tor Vergata, Viale Montpellier 1, 00133 Rome, Italy; 3Pathology Unit, Azienda Unità Sanitaria Locale-Istituto di Ricovero e Cura a Carattere Scientifico di Reggio Emilia, 42122 Reggio Emilia, Italy; mariapaola.bonasoni@ausl.re.it; 4Department of Foetal and Maternal Pathology, Sant’Anna Hospital, 10126 Turin, Italy; giovanni.botta@unito.it; 5Histopathology and Molecular Diagnostics, Careggi University Hospital, 50134 Florence, Italy; castiglionef@aou-careggi.toscana.it; 6Department of Public Health, University of Naples Federico II, 80131 Naples, Italy; maria.darmiento@unina.it; 7Fetal-Perinatal Pathology Unit, Istituto di Ricovero e Cura a Carattere Scientifico-Istituto Giannina Gaslini, 16147 Genoa, Italy; eziofulcheri@gaslini.org; 8Division of Anatomic Pathology, Department of Surgical and Diagnostic Sciences (DISC), University of Genova, 16132 Genova, Italy; 9Pathology Unit, ASST Fatebenefratelli Sacco, Department of Biological and Clinical Sciences, University of Milan, 20157 Milan, Italy; manuela.nebuloni@unimi.it

**Keywords:** SARS-CoV-2, placental histopathology, placental inflammation

## Abstract

The variety of placental morphological findings with SARS-CoV-2 maternal infections has raised the issue of poor agreement in histopathological evaluation. The aims of this study were: to describe the histopathological placental features of a large sample of SARS-CoV-2-positive women who gave birth in Italy during the COVID-19 pandemic, to analyse the factors underlying these lesions, and to analyse the impact of placental impairment on perinatal outcomes. From 25 February 2020 to 30 June 2021, experienced perinatal pathologists examined 975 placentas of SARS-CoV-2-positive mothers enrolled in a national prospective study, adopting the Amsterdam Consensus Statement protocol. The main results included the absence of specific pathological findings for SARS-CoV-2 infections, even though a high proportion of placentas showed signs of inflammation, possibly related to a cytokine storm induced by the virus, without significant perinatal consequences. Further research is needed to better define the clinical implications of placental morphology in SARS-CoV-2 infections, but the results of this large cohort suggest that placentas do not seem to be a preferential target for the new Coronavirus infection.

## 1. Introduction

Since its outbreak in late December 2019, the Severe Acute Respiratory Syndrome Coronavirus 2 (SARS-CoV-2) infection has represented a significant challenge for worldwide sanitary systems because of its high death rate and rapidity of transmission.

A two-wave pattern of the Coronavirus Disease 2019 (COVID-19) during the 2020 pandemic was observed in Italy, with the first wave during spring 2020 followed by the second one in autumn and extended until the end of June 2021.

From 28 December 2020 to 19 May 2021, the Italian National COVID-19 Integrated Surveillance System reported 2,083,674 cases of COVID-19 infections, of which 23,170 were genotyped with sequencing. The Wild-type virus was predominant until January 2021, and the national prevalence of the VOC-202012/01 B.1.1.7 lineage (the Alpha variant) in February and March 2021 was 54.0% and 86.7%, respectively. In June 2021, the number of cases related to the B.1.617.2 lineage (the Delta variant) in Italy exceeded those related to the Alpha variant [1].

Regarding COVID-19 infections in pregnancy, international literature suggests that the Alpha and Delta variants had a worse impact on maternal and perinatal outcomes [2,3]. From the beginning of the pandemic, the Italian Obstetric Surveillance System (ItOSS) launched a national population-based prospective study [4], enrolling 5734 pregnant women with a confirmed SARS-CoV-2 infection that were admitted to a hospital until 30 June 2021. Key findings include 64.3% of asymptomatic women, 12.8% of COVID-19 pneumonia, 11.1% of preterm births, 3.3% of women with a severe COVID-19 disease, and the worst maternal and perinatal morbidity associated with the Alpha variant compared to the Wild-type virus. Stillbirths (0.7%) and early neonatal deaths (0.2%) were neither associated with maternal pneumonia nor with the different viral strains.

The high number of women who contracted a SARS-CoV-2 infection during pregnancy has raised several questions on the physiopathology of mother-to-child transmissions and on the role of the placenta in this setting. The hypothesis of COVID-19 transplacental transmission remains controversial: SARS-CoV-2 RNA has been detected through the PCR technique, but a low tissue viral load found in the placentas was unable to induce an inflammatory response, as confirmed by maternal and neonatal clinical outcomes, placental histology, and molecular data that did not differ in the case of a SARS-CoV-2 infection [5]. These results suggest that the placenta may not be a preferential target for the new Coronavirus infection, possibly due to the minimal expression in the placenta of canonical cell-entry mediators for the virus. In the case of a significant placental viral load, although there is an absence of a vertical transmission, high placental viremia may determine a functional impairment of the organ, possibly affecting the neonatal clinical outcome [6].

The variety of morphological scenarios of third-trimester placentas with maternal infections of SARS-CoV-2 has highlighted the problem of diagnosis heterogeneity and poor agreement in histopathological evaluation. Several scientific publications suggested the existence of typical microscopic features related to maternal and foetal malperfusion, refuted by subsequent studies [7,8]. The histopathological triad most frequently encountered includes chronic histiocytic intervillositis (CHI), villous trophoblast necrosis, and intervillous fibrin deposition (IFD), the so-called “SARS-CoV-2 placentitis” [9].

The primary aim of this study is to describe the histopathological placental features of a large sample of SARS-CoV-2-positive women who gave birth in Italy from February 2020 to June 2021. Secondary aims are to describe the factors associated with the SARS-CoV-2 histopathological placental lesions and the impact of placental impairment on perinatal outcomes.

## 2. Materials and Methods

### 2.1. Study Population

The study population comprised a sample of women with a current or previously confirmed SARS-CoV-2 infection during pregnancy that were admitted to a hospital for childbirth from 25 February 2020 to 30 June 2021 in seven Italian Regions (Piedmont, Lombardy, Liguria, Emilia-Romagna, Tuscany, and Campania) and in the Autonomous Province of Trento, which together covers 49% of Italian births. The study was part of the national prospective population-based study on SARS-CoV-2 infections in pregnancy, at birth, and post-partum, coordinated by the ItOSS surveillance system [4]. A confirmed SARS-CoV-2 infection was defined as the detection of viral RNA on reverse transcriptase-polymerase chain reaction (RT-PCR) testing of a nasopharyngeal swab, both for mothers and newborns.

### 2.2. Data Collection Procedures

Reference clinicians from the 143 maternity units (Appendix A) in the participating regions collected the placenta, membranes, and umbilical cord of any woman who gave informed consent to participate in the study. For each case, an identification number was generated, and clinicians received a link to the LimeSurvey GmbH platform where they entered information about maternal socio-demographic characteristics, medical and obstetric history, disease management, mode of delivery, and maternal and perinatal outcomes.

Each participating region identified a reference laboratory with expert pathologists in the perinatal field. The Pathology Unit of Sacco Hospital (Milan) collected the specimens for both the Lombardy Region and the Autonomous Province of Trento; Città della Salute e della Scienza di Torino (Turin) for the Piedmont Region; IRCCS Istituto Giannina Gaslini (Genoa) for the Liguria Region; Azienda USL-IRCCS di Reggio Emilia (Reggio Emilia) for the Emilia-Romagna Region; Careggi Hospital (Florence) for the Tuscany Region; and University Federico II of Naples (Naples) for the Campania Region.

A panel of six national experienced perinatal pathologists was appointed to develop: (i) a standardised protocol for the biological samples collection, storage, and transport procedures (Appendix A), (ii) an online data collection form (Appendix A), and (iii) a standardised protocol for the macroscopic and microscopic biological samples description according to the guidelines of the Amsterdam Consensus [10]. All reference laboratories approved and adopted the shared protocol. For each case for which the clinicians were able to collect the biological samples, a link to LimeSurvey GmbH was created, through which the reference pathologists entered the histopathological data in the online form where a selection of maternal and neonatal clinical information were preloaded. Both clinical and histological data were uploaded to a secure server of the Italian National Institute of Health.

### 2.3. Sociodemographic and Clinical Characteristics of the Mothers

The following variables were adopted to describe the mother’s socio-demographic and clinical conditions: age (<30, 30–34, and ≥35 years); citizenship (Italian and non-Italian); gestational age at SARS-CoV-2 diagnosis (<14 weeks, 14–27 weeks, and ≥28 weeks); obesity (body mass index—BMI ≥30 Kg/m^2^); multiple pregnancies; previous comorbidities and pregnancy complications (diabetes; autoimmune diseases; preeclampsia); COVID-19 pneumonia; time between positive swab and delivery (≤14 days and >14 days); intensive care unit (ICU) admission; mechanical ventilation support (non-invasive mechanical ventilation or intubation or extracorporeal membrane oxygenation—ECMO); and maternal death.

The following variables were adopted to describe the newborns’ clinical conditions: gestational age at birth (≥31, 32–36, and ≥37 weeks of gestation); Apgar score at 5′ (<7 and ≥7); small for gestational age (SGA) (birthweight <10th centile according to the Italian Neonatal Study charts [11]); weight at birth (<1500 g, 1500–2499 g, and ≥2500 g); SARS-CoV-2-positive swab at birth; admission to neonatal intensive care unit (NICU); stillbirth; and neonatal death.

### 2.4. Biological Samples

In accordance with the standardised protocol, the placentas, membranes, and umbilical cords were stored in hermetic containers in 10% formalin and at room temperature in order to provide immediate fixation after birth. The containers were labelled using the case identification number and were sent to the reference laboratories. The biological samples were macroscopically and microscopically analysed and described according to the Amsterdam Placental Workshop Group Consensus Statement [10].

The macroscopic analysis included the placental, umbilical cord, and membranes’ morphological features. Information regarding the cord, length, diameter, coiling, and type of insertion was recorded. The membranes were described according to colour and insertion. The placental trimmed weight, dimensions, chorionic vessel anomalies, and parenchymal lesions (infarcts, hematoma) were described.

Microscopic features included the detection of inflammation such as funisitis (F), chorioamnionitis (CA), villitis, and intervillitis and the grading of their severity. Maternal and foetal vascular perfusion were also analysed, evaluating fibrin deposition, intervillous haemorrhage, the location of thrombohematoma, and infarcts.

The histopathologic features adopted to describe the severity of the placental SARS-CoV-2 infection were defined as follows:

Absence of inflammation: no intervillous fibrin, no acute, chronic, or mixed villitis and intervillitis;

Mild inflammation: mild intervillous fibrin and/or mild acute, chronic, or mixed villitis and/or intervillitis;

Moderate or severe inflammation: moderate or severe intervillous fibrin and/or moderate or severe acute, chronic, or mixed villitis and/or intervillitis; the inflammation was classified as severe when all three conditions were present.

### 2.5. Statistical Analysis

Percentage distributions of the placentas by socio-demographic and clinical characteristics of the women were calculated. The association between these variables and the severity of the SARS-CoV-2 inflammation of the placenta was assessed through a multinomial logistic regression model directed to estimate the mutually adjusted odds ratios (ORs) and 95% confidence interval (CI). The macroscopic and microscopic characteristics of the placentas were described through percentage distributions and prevalence. Bivariate tables and Pearson’s chi-squared test were used to assess the association between the presence of the inflammation of the placenta and the presence of other histopathological features indicative of a SARS-CoV-2 infection (intervillar haemorrhage, CA, and F). Logistic regression models were used to assess the association of placental inflammation, CA, and F with some perinatal outcomes through the estimation of ORs and 95% CI adjusted for maternal socio-demographic, obstetrics, and medical characteristics. Statistical analyses were performed using the statistical package STATA/MP 14.2.

### 2.6. Ethics

The study was approved by the Ethics Committee of the *Istituto Superiore di Sanità* (Italian National Institute of Health) (protocol 0010482 CE 01.00, Rome, 24 March 2020). The study protocol can be retrieved at https://www.epicentro.iss.it/en/coronavirus/sars-cov-2-pregnancy-childbirthbreastfeeding-prospective-study-itoss (in Italian, accessed on 15 June 2022). Every woman willing to participate signed an informed consent form as well as the father of the newborn to authorise the examination of the foetal annexes.

## 3. Results

### 3.1. Characteristics of the Study Population

Table 1 describes the socio-demographic and obstetric characteristics, as well as the main maternal and neonatal outcomes of the cases enrolled in the ItOSS study for which biological samples were available. The great majority of the enrolled mothers were diagnosed as positive to SARS-CoV-2 during the third trimester of pregnancy (81.7%), 14.4% during the second, and 3.9% during the first. More than half of the women (53.3%) were diagnosed as positive within 14 days of delivery.

From February 2020 to June 2021, 975 placentas of SARS-CoV-2-positive women who gave birth within the ItOSS cohort were collected and sent to the regional reference laboratories. As required by the study protocol, almost all of them (94.0%) were immediately fixed in 10% formalin and were stored in hermetic containers at room temperature; 5.7% were stored freshly and 0.3% were vacuum packed (Appendix A).

### 3.2. Macroscopic Examination

On macroscopic examination, 98.6% of the placentas were single, with the cord inserted centrally in 63.1%, marginally in 28.3%, and with velamentous insertion in 1.0% of cases; information on the insertion was missing in 7.6% of cases (Appendix A). The majority of the placentas (66.6%) had a weight between the 10th and 90th centile, 9.3% < 10th centile, and 4.6% > 90th centile; for 19.5%, the information on weight was not reported. Strictures, knots, and twists of the cord were present in 2.1%, 6.0%, and 1.0% of cases, respectively. The membranes were described as thin (44.2%), complete (24.6%), incomplete (19.1%), and thickened (5.3%); the description was not reported in 6.8% of cases.

### 3.3. Microscopic Examination

A microscopic examination was performed according to the main histopathologic lesions reported in SARS-CoV-2 inflammatory and non-inflammatory findings, following the Amsterdam Consensus [10].

Inflammatory lesions included CA, F, villitis of unknown aetiology (VUE), CHI, and IFD.

Non-inflammatory lesions comprised intervillar haemorrhages (recent, organising, and organised), maternal vascular malperfusion (MVM) with infarcts, abnormal decidual arteries, foetal vascular malperfusion (FVM) with vascular thrombosis, and avascular villi.

Placental maturity was evaluated as immature, dysmature, and hypermature.

CA and F were classified in mild, moderate, and severe according to stage and grade [12].

CA was present in 40.4% (*n* = 394) of the analysed membranes, 22.9% being mild and 17.5% moderate/severe (Appendix A). F was identified in 18.2% (*n* = 177) of the cords, divided into 14.1% mild and 4.1% severe, respectively (Appendix A).

Further analysing these results, the presence of CA and F considered separately was observed in 227 cases (23.3%) and 10 (1.0%), respectively, and both CA and F were found in 167 cases (17.1%). Among the 404 placentas characterised by the presence of CA and/or F, 170 (42.3%) women were diagnosed as SARS-CoV-2 positive within 7 days of delivery (data not shown).

We evaluated the inflammatory status of the chorionic disc including IFD and inflammatory infiltrate within the stroma of terminal villi and/or into the intervillous space. The latter was mainly composed of lymphocytes and histiocytes. According to these features, 37.4% (*n* = 365) of the placentas did not show inflammation, 32.3% (*n* = 315) showed mild inflammation, and 30.2% (*n* = 295) presented a moderate or severe condition. In this group, 279 cases were classified as moderate and 16 as severe (Table 1). The 16 cases identified as severe showed CHI and massive IFD reported as typical lesions of SARS-CoV-2 in the placenta, although unusual.

VUE was observed in 186 cases (19.1%), with a prevalence of lymphohistiocytic infiltrate and rarely plasma cells (*n* = 2; 1.1%) (data not shown).

Table 2 shows the ORs of developing mild and moderate/severe inflammation comparing the inflammation severity with cases without inflammation. The occurrence of a moderate/severe inflammation was higher among women of non-Italian citizenship (OR 1.82, 95% CI 1.23–2.70) and among those who contracted a SARS-CoV-2 infection during the Alpha variant period, although at limits of statistical significance (OR 1.45, 95% CI 0.98–2.16). In addition, the Alpha variant period was also associated with a higher occurrence of mild inflammation (OR 1.59, 95% CI 1.08–2.33). Four cases of preeclampsia (two associated with mild inflammation and two with moderate/severe inflammation) were excluded from the model due to the small number.

A further analysis was directed to compare the prevalence and severity of placental inflammation in the presence or absence of maternal COVID-19 pneumonia. No statistically significant difference was observed for CA, F, chorionic disc inflammation, IFD, and inflammatory infiltrate (Appendix A).

Intervillar haemorrhage affected 31.9% of the placentas: 15.5% as recent haemorrhages, 11.8% in organisation, 4.3% organised, and 0.3% pseudocystic (Appendix A). The multivariate analysis detected a statistically significant association between the mother’s BMI ≥ 30 Kg/m^2^ and intervillar haemorrhage occurrence (OR 1.53, 95% CI 1.02–2.28) (data not shown). The occurrence of intervillar haemorrhages and mild and moderate/severe CA and F was higher in the placentas with detected SARS-CoV-2 inflammatory signs (*p* < 0.001) (Appendix A). Table 3 describes the impact of the inflammation of the placenta, CA, and F on a selection of adverse neonatal outcomes (SGA, preterm birth, and NICU admission) after adjusting for maternal characteristics. No statistically significant association was detected.

Placental infarcts in MVM were found in 192 samples (19.8%), further subdivided in recent (*n* = 65; 6.7%), organising (*n* = 64; 6.6%), and old (*n* = 63; 6.5%) (Appendix A).

The decidual bed was affected in 159 cases (16.3%) with no physiological transformation of the maternal arterioles (data not shown). Among these cases, the incidence of diabetes (four cases), hypertension (one case), and pre-eclampsia (one case) was low. Decidual atherosis was detected in 38 cases, one associated with diabetes. On the whole, decidual vessels were abnormal in 197 placentas (20.2%) and foetal growth restriction was found in 44 cases (4.5%) (data not shown).

FVM was analysed including cases with vascular thrombosis in the chorionic, stem, and intermediate vessels. These findings were present in 21 cases (2.2%) and avascular villi were detected in eight of them. Overall, avascular villi were found in 139 samples (14.3%) (data not shown). Therefore, FVM was observed in 152 placentas (15.6%).

Regarding placental maturation, most of the placentas (72.5%) were coherent with gestational age, 8.1% immature, 5.1% hypermature, and 14.3% dysmature (Appendix A).

## 4. Discussion

The effects of SARS-CoV-2 in pregnancy have been currently updated, including maternal and neonatal outcomes and placental pathological findings, but informative data are still lacking. Large studies, which thoroughly analyse maternal clinical information, placental findings, and neonatal outcome, have yet to be reported, and the current available information is based on reviews and meta-analyses [13,14,15,16,17]. Herein, we present a series of 975 placentas of SARS-CoV-2-positive women analysed in six different regional Italian reference laboratories. The study was conducted from February 2020 to June 2021. In order to have comparable and statistically useful data, all the placentas were assessed macroscopically and microscopically through a standardised protocol [10]. To date, only two other single studies considered a relevant population with 101 and 187 cases, respectively, including data from women, placentas, and newborns [18,19]. Only an extensive meta-analysis is available with 1452 cases retrieved from 30 publications [16]. Other two recent series, although less consistent in the number of cases, will be taken into account for comparison [19,20].

### 4.1. Main Findings

In our study, most of the pregnant women were diagnosed as positive to SARS-CoV-2 during the third trimester of pregnancy (81.7%), of which 53.3% were diagnosed within 14 days of delivery; 14.4% contracted the virus during the second trimester and 3.9% during the first trimester.

Regarding the gross findings, 98.6% of the placentas were single, and the cord insertion was mainly reported as centrally located (63.1%), with fewer cases reported as marginal (28.3%) and in 1.0% as velamentous (information was missing in 7.6% of cases). Strictures, notes, and abnormal coiling were also observed in a few cases. However, it is highly unlikely that SARS-CoV-2 in the third trimester affected cord development and twisting. Instead, in most of the studies [16,19,20], this information was not mentioned, and only the weight centile was assessed. In our population, most of the placentas presented a weight between the 10th and 90th centile (66.6%), with only 9.3% below the 10th centile and 4.6% above the 90th (weight was not reported for one in five cases). In the already mentioned meta-analysis [16], small placentas were recorded in 87/585 (15%) cases and big specimens were reported in 61/495 (12%) cases. In the series by Stenton et al. [20], 47/58 (81%) placentas had a normal weight, only 8/58 (13.8%) were small, and 3/58 (5.2%) were big for gestation, respectively. On the whole, including also our cases, this data suggests that SARS-CoV-2 does not significantly affect placental weight.

As far as the inflammatory lesions were concerned, CA was found in 40.4% of the specimens (394), further subdivided into mild (22.9%) and moderate/severe (17.5%). Funisitis was also notably present in 177 cases (18.2%), with 14.1% being mild and 4.1% being moderate/severe. On the whole, inflammation of the membranes and/or the cord was present in 404 placentas, almost half of the specimens analysed (41.4%). This percentage was significantly higher than the one reported by Suhern et al. [16], in which the combined inflammation of the membranes and cord was 26%, considering 376/1.430. The CA finding was rarely detected in the study by Stenton et al. [20], with one case found in the 61 placentas examined, as well as in the series by Argueta et al. [19], with two cases found in 20 samples. Instead, in our population, acute inflammation was a significant finding and involved both maternal and foetal responses. Moreover, most of the placentas showing CA and/or F (170/404; 42.3%) involved women with a diagnosis of SARS-CoV-2 within 7 days of delivery. This result may suggest a possible involvement of SARS-CoV-2, considering the cytokine storm that this virus may induce, such as tumour necrosis factor (TNFa), interferon γ (INFγ), interleukin-6 (IL6), interleukin- 1b (IL1b), and chemokines such as monocyte chemoattractant protein-1 (MCP-1/CCL2), which can also favour granulocyte stimulation [21,22,23]. CA is expected to occur around 10–20% of the time in non-COVID pregnancies [24]. In addition, maternal hypoxia and systemic inflammation may be responsible for the altered placental amniotic fluid barrier and possible co-infection [16]. However, cytokine expression in pregnant women with a SARS-CoV-2 infection is yet to be fully understood. In vitro studies detected different cellular immune reactions to SARS-CoV-2 and cytokine secretion in pregnant women compared to non-pregnant women. Especially IL-8 concentration, which has a chemotactic effect on neutrophils and is increased in SARS-CoV-2-positive women [25].

On the whole, the inflammatory condition of the placental parenchyma showed that 62.5% of cases (*n* = 610) presented mild, moderate, or severe lymphohistiocytic infiltrate within or close to the villi, and IFD. Only 16 cases, 1.6%, presented CHI and IFD. In the meta-analysis by Suhern et al. [16], CHI and IFD were low with 57 cases out of 1096 (5%) and with 13/924 samples (1%), respectively. In the series by Stenton et al. [20], the concomitant presence of CHI and IFD was found in 48/61 placentas (79%). Argueta et al. reported CHI and IFD only in 2/20 placentas (10%) [19]. However, these series were smaller compared to our population and the first one included stillbirths and different gestational ages. Considering our results and the meta-analysis by Suhern [16], the incidence of this unusual lesion was overall low. CHI associated with IFD and trophoblast necrosis was defined as “SARS-CoV-2 placentitis”. This lesion may be a risk factor for miscarriage or stillbirth and is not necessarily associated with symptomatic mothers or vertical transmission to the newborn, despite COVID-19 tissue positivity [9,19,20]. The exact pathophysiological mechanism behind this lesion has yet to be established. A possible explanation may rely on the combined effect of the dysregulation of the immune response and a procoagulant status in the mother and/or the foetus, which can activate a CHI, IFD, and trophoblast necrosis cascade [20]. In the pre-pandemic era, the association between CHI and massive IFD was rarely described, although CHI was known to have an immunological basis and a high risk of recurrence [26,27]. Moreover, CHI was associated with intrauterine foetal death (IUFD) and early onset intrauterine growth restriction (IUGR). IFD, when massive, is also known as massive perivillous fibrin deposition (MPFD) or maternal floor infarct (MFI). In general, IFD can significantly obstruct maternal blood flow with adverse foetal consequences such as renal dysplasia, stillbirth, IUGR, preterm birth, and neonatal neurological impairment [27]. However, in our series, there was no significant association with SGA neonates and/or NICU admission. Even Stenton did not find substantial neonatal morbidities [20]. The so-called “SARS-CoV-2 placentitis” and how it may affect the foetus is still unclear. Probably the timing of maternal viral infection may be a factor in explaining newborns’ outcome.

VUE was observed in 186 cases (19.1%), with a prevalence of lymphohistiocytic infiltrate and rarely plasma cells (*n* = 2; 1.1%). In the series by Suhern et al. [16], VUE was detected in 187/1177 (16%) of cases, a slightly lower percentage compared to our population. Argueta found VUE in 4/20 samples (20%), and Stenton only in 2/61 (3.2%) [20]. However, the last group was not homogeneous for gestational age and foetal outcome including stillbirths. On the whole, VUE seems to be a common finding in placentas with a SARS-CoV-2 infection compared to normal pregnancies, in which the occurrence is below 10% [28]. This is currently considered as a dysregulation of the maternal immune system due to the possible alteration of lymphocyte and macrophage function in the SARS-CoV-2 infection [29].

We found a high percentage of placentas (31.9%) with intervillar haemorrhage in different stage of organisation. In the other case series, this feature was not singularly considered. Intervillous thrombi were mentioned by Glynn (3/90 placentas; 3.3%) [30] and Argueta (1/20 samples; 5%) [19]. However, considering only the organised haemorrhages, also defined as intervillous thrombi, the percentage was similar (4.3%) to other reports [19,30]. Although IVT occurrence is higher in diabetes and SGA newborns [31,32], we detected a low incidence of FGR (*n* = 44; 4.5%) and diabetes (*n* = 5; 0.5%).

Placental infarcts in different stages of organisation were observed in 192 samples (19.8%) compared to 101/708 cases (14%) retrieved by the meta-analysis by Suhern et al. [16]. MVM included 197 samples (20.2%) with abnormal decidual arteries with no physiological transformation and acute atherosis. Despite the typical occurrence in pre-eclampsia and hypertension, in our population only two mothers were affected [33]. Glynn et al. found 27/90 cases (30%) with MVM and, similar to our cohort, no increased frequency of hypertension, pre-eclampsia, or foetal growth restriction [30]. Even in the review by Sharps et al., MVM occurrence in SARS-CoV-2 maternal infections was not different from the controls [34].

FVM was found in a total of 152 (15.6%) samples, combining vascular thrombosis and avascular villi.

Regarding FVM, Hessami et al. specifically analysed 10 studies [35], but no significant statistical difference was found between SARS-CoV-2 mothers and negative placentas. In another study by Ramey-Collier et al., FVM was detected in 1/112 placentas (0.9%) [36]. Instead, Glynn et al. observed a high percentage of FVM, 26/90 cases (28%) [30]. This discrepancy is difficult to explain with the current knowledge, and wider series would be necessary in order to better assess this finding and potential foetal consequences.

Maturation disorders of the placenta included dysmaturity in 135 cases (13.8%), but only two mothers presented diabetes. Our findings were similar to what was reviewed by Suhern et al. with 94/674 cases (14%) [16]. Tasca et al. found delayed villous maturation more frequently in symptomatic patients that were pharmacologically treated [37], approaching a statistical difference, but the total group examined was small (*n* = 64).

### 4.2. Clinical Implications

Overall, grossly and microscopically, the placentas did not show a specific pattern of pathological findings for SARS-CoV-2, neither in women with a mild/moderate COVID-19 disease nor in those with a severe disease. Although 81.7% of the infections occurred in the third trimester, making the development of placental lesions unlikely, the absence of pathognomonic signs of severe placental compromise even in the cases diagnosed early in pregnancy seemed coherent with the stability of stillbirths and early neonatal death rates observed in Italy during the pandemic period compared to previous years [4].

In our series, the high percentage of placentas with signs of CA and/or F (42.3%) may be interpreted as an increased expression of chemokines, cytokines, and inflammatory-related genes due to the SARS-CoV-2 infection [19]. Although this hypothesis suggests a possible involvement of the virus in the higher rate of CA, the impact of the inflammation of the placenta, CA, and F after adjustment for maternal characteristics did not show any impact on SGA, preterm birth, and NICU admissions. Moreover, the 20.2% of MVM detected in the placentas mismatched with the rare clinical diagnosis of hypertension, pre-eclampsia, and foetal growth restriction known to be related to the MVM non-inflammatory lesion [35]. Interestingly, women of non-Italian citizenship and those who contracted the infection during the Alpha period showed a higher risk of both severe COVID-19 pneumonia and moderate/severe inflammation of the placenta compared to Italian mothers and to those who contracted the infection during the Wild-type period. Women with a BMI ≥ 30 Kg/m^2^ had a higher risk of COVID-19 pneumonia and occurrence of intervillar haemorrhage compared to non-obese women.

Further research is needed to better define the clinical implications of a placenta’s morphology due to a SARS-CoV-2 infection but the results of this large cohort suggest the absence of pathognomonic placental lesions for SARS-CoV-2 infections during pregnancy, corresponding to defined perinatal outcomes.

### 4.3. Strengths and Limitations

The participation of seven Italian regions covering 49% of national births, the large size of the present series, the confirmed SARS-CoV-2 infection through the RT-PCR testing of nasopharyngeal swabs for both mothers and newborns, and the selection of reference laboratories with expert pathologists in the perinatal field adopting a shared protocol referring to the Amsterdam Consensus Statement [10] are important strengths of the present study. The availability of prospective epidemiological data describing the socio-demographic and clinical characteristics of the women represents a further strength, which facilitated the correct interpretation of the data. The study limitations include the subnational series, although a lack of representativeness is unlikely given that the seven participating regions covered all geographical areas of the country, and the unavailability of a control group of placentas.

## 5. Conclusions

To the best of our knowledge, this is the largest series of placentas (N = 975) macroscopically and microscopically homogenously analysed according to Amsterdam Consensus guidelines [10]. Compared to the most recent literature, the placentas under consideration did not present a specific pattern of pathological findings for SARS-CoV-2 and did not seem to be a preferential target for the new Coronavirus infection. The detected high incidence of signs of inflammation, including CA, F, IFD, and VUE could be attributed to a possible immune response induced by the virus, although the effect of SARS-CoV-2 in the placenta is still unclear and under investigation.

## Figures and Tables

**Table 1 microorganisms-10-01435-t001:** Maternal and perinatal characteristics.

Maternal Characteristic	N = 975	
	*n*	%
Age (12 missing)		
<30 years	284	29.5
30–34 years	336	34.9
≥35 years	343	35.6
Citizenship		
Italian	760	77.9
Non-Italian	215	22.1
Previous comorbidities (14 missing)		
No	825	85.8
Yes	136	14.2
Pre-gestational diabetes	15	1.6
Autoimmune diseases	31	3.2
BMI ≥ 30 Kg/m^2^ (8 missing)		
No	841	87.0
Yes	126	13.0
COVID-19 pneumonia		
No	899	92.2
Yes	76	7.8
Multiple pregnancies		
No	960	98.5
Yes	15	1.5
Gestational trimester at SARS-CoV-2 diagnosis (7 missing)		
I (<14 weeks)	38	3.9
II (14–27 weeks)	139	14.4
III (≥28 weeks)	791	81.7
Maternal outcomes		
Severe morbidity	17	1.7
Intensive care unit admission	19	1.9
Mechanical ventilation support *	21	2.2
Death	1	0.1
Placental disc inflammation **		
Absent	365	37.4
Mild	315	32.3
Moderate	279	28.6
Severe	16	1.6
**Perinatal outcomes**	**N = 990**	
	* **n** *	**%**
Stillbirths	5	0.5
Livebirths	985	99.5
Neonatal deaths	4	0.4
NICU admission	75	7.6
Weight at birth (11 missing)		
<1500 g	21	2.2
1500–2499 g	66	6.8
≥2500 g	887	91.1
Apgar <7 at 5′ (39 missing)	12	1.3
SARS-CoV-2-positive swab at birth	13	1.3

BMI: body mass index; COVID-19: Coronavirus Disease 2019; NICU: neonatal intensive care unit; SARS-CoV-2: Severe Acute Respiratory Syndrome Coronavirus 2. * Non-invasive mechanical ventilation or intubation or extracorporeal membrane oxygenation (ECMO). ** Absent inflammation: no intervillous fibrin, no acute, chronic, or mixed villitis and intervillitis; mild inflammation: mild intervillous fibrin and/or mild acute, chronic, or mixed villitis and/or intervillitis; moderate or severe inflammation: moderate or severe intervillous fibrin and/or moderate or severe acute, chronic, or mixed villitis and/or intervillitis; the inflammation was classified as severe when all three conditions were present.

**Table 2 microorganisms-10-01435-t002:** Odds ratios of mild and moderate/severe placental disc inflammation— multinomial logistic regression model (*n* = 932, four cases of preeclampsia excluded).

Variable	Mild Inflammation vs. No Inflammation			Moderate/Severe Inflammation vs. No Inflammation		
	OR	95% CI		OR	95% CI	
SARS-CoV-2 type						
Wild type	1			1		
Alpha variant	1.59	1.08	2.33	1.45	0.98	2.16
Maternal age						
<30 years	1			1		
30–34 years	0.96	0.64	1.43	0.95	0.63	1.41
≥35 years	1.10	0.74	1.63	0.86	0.58	1.30
Citizenship						
Italian	1			1		
Not Italian	1.11	0.73	1.67	1.82	1.23	2.70
Gestational trimester at SARS-CoV-2 diagnosis						
I (<14 weeks)	1			1		
II (14–27 weeks)	1.72	0.69	4.29	1.82	0.72	4.60
III (≥28 weeks)	1.88	0.79	4.50	1.93	0.80	4.65
COVID-19 pneumonia						
No	1			1		
Yes	0.85	0.46	1.56	1.05	0.58	1.91
Interval between diagnosis and delivery						
≤14 days	1			1		
>14 days	0.85	0.59	1.22	0.86	0.59	1.25
BMI ≥ 30 kg/m^2^						
No	1			1		
Yes	1.28	0.80	2.04	1.12	0.69	1.81
Pre-gestational diabetes						
No						
Yes	1.44	0.31	6.65	3.39	0.86	13.32
Autoimmune disease						
No						
Yes	1.08	0.45	2.61	1.04	0.41	2.59

BMI: body mass index; COVID-19: Coronavirus Disease 2019; SARS-CoV-2: Severe Acute Respiratory Syndrome Coronavirus 2. * No inflammation: no intervillous fibrin, no acute, chronic, or mixed villitis and intervillitis; mild inflammation: mild intervillous fibrin and/or mild acute, chronic, or mixed villitis and/or intervillitis; moderate or severe inflammation: moderate or severe intervillous fibrin and/or moderate or severe acute, chronic, or mixed villitis and/or intervillitis; the inflammation was classified as severe when all three conditions were present.

**Table 3 microorganisms-10-01435-t003:** Odds ratios of the occurrence of preterm births, small for gestational age, and neonatal intensive care unit admission—logistic regression models.

Variable	Preterm Birth Yes vs. No			SGA Yes vs. No			NICU Admission Yes vs. No		
	OR	95% CI		OR	95% CI		OR	95% CI	
Model 1 *, placental disc inflammation **									
Absent	1			1			1		
Mild	0.62	0.33	1.16	0.88	0.52	1.50	1.25	0.66	2.35
Moderate/severe	0.92	0.51	1.65	0.99	0.58	1.70	0.95	0.48	1.87
Model 2 *, chorionamnionitis									
Absent	1			1			1		
Mild	0.60	0.31	1.15	0.76	0.43	1.35	0.63	0.30	1.29
Moderate/severe	0.55	0.25	1.19	0.82	0.44	1.53	1.14	0.56	2.32
Model 3 *, funisitis									
Absent	1			1			1		
Mild	0.45	0.18	1.12	0.60	0.28	1.28	1.01	0.46	2.18
Moderate/severe	0.71	0.15	3.41	1.19	0.40	3.52	1.81	0.49	6.67

* ORs adjusted for maternal age, citizenship, gestational trimester of diagnosis, COVID-19 pneumonia, obesity, comorbidities, and interval between diagnosis and delivery. NICU: neonatal intensive care unit; SGA: small for gestational age. ** Absent inflammation: no intervillous fibrin, no acute, chronic, or mixed villitis and intervillitis; mild inflammation: mild intervillous fibrin and/or mild acute, chronic, or mixed villitis and/or intervillitis; moderate or severe inflammation: moderate or severe intervillous fibrin and/or moderate or severe acute, chronic, or mixed villitis and/or intervillitis; the inflammation was classified as severe when all three conditions were present.

## Data Availability

The data presented in this study are available on request from the corresponding author.

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
