# Peer review of "Placental Characteristics of a Large Italian Cohort of SARS-CoV-2-Positive Pregnant Women"

_microorganisms, 2022, doi:10.3390/microorganisms10071435_

Round 1

Reviewer 1 Report

The study by Salvatore et al investigated the placental characteristics of the Italian cohort of SARS-CoV-2 positive pregnant women and suggested that placentas seem not to to be a preferential target for the novel Coronavirus infection. This study is well written and presented the detailed results of microscopic and macroscopic characteristics of placenta. This study will helps scientific community to understand the impact of current COVID-19 pandemic in pregnancy and fetal outcomes. 

Author Response

We warmly thank the reviewer for their appreciation of our work.

Reviewer 2 Report

In the present work, Salvatore and collaborators do excellent epidemiological work concerning pathological evidence in extra-embryonic tissues of women who experienced SARS-CoV-2 infection. However, the effect of perinatal infection of this new coronavirus is a widely controversial issue. Various studies suggest a low vertical transmission, but the affection of the placenta produced by a collateral effect of this infection highlights the importance of the findings in this study. Remember that groups have determined a null expression of the canonical receptors for the access of the virus to the cellular components. It is clear that other molecules that can act as receptors for the virus have also been reported, in addition to the existence of different mechanisms that could damage the placental barrier to affect this structure. 

The findings in this work suggest that in the population studied. They did not find any effect of those already reported in the tissues of pregnant women who experienced a SARS-CoV-2 infection. However, the authors discuss that the identified damage is associated with a collateral effect due to the systemic inflammation generated by the disease; however, the cytokine storm may be more associated with patients who experienced severe covid. Therefore, I believe that if one of your hypotheses is related to this inflammation in pregnant women, it would be convenient to demonstrate cytokine levels in your population, especially in those patients who did not develop severe covid.

Finally, I suggest you qualify your conclusions and your statements in the introduction. The presence of putative receptors for the virus cannot be just the conventional mechanisms viruses can use to gain access. There is also evidence of the expression of these receptors in extra-embryonic tissues. The cytokine storm is an effect identified in the population that experiences severe covid, the viremia, and these cytokines are factors that could generate these pathological phenomena in the placenta. However, it is crucial to handle it as a possibility. There are reports that abortive infections, infections that do not produce viral particles, can generate or trigger an inflammatory immune response. So it should not be forgotten that although some studies have identified infection in the placentas, this effect may be possible without the need to transmit the virus to the fetus.

Author Response

We kindly appreciate your observations. Regarding these statements: “Remember that groups have determined a null expression of the canonical receptors for the access of the virus to the cellular components. It is clear that other molecules that can act as receptors for the virus have also been reported, in addition to the existence of different mechanisms that could damage the placental barrier to affect this structure.“

As far as the “cytokine storm” observation is concerned, we further examined differences in prevalence and severity of placental inflammatory status in presence or absence of maternal COVID-19 pneumonia. However, no statistically significant difference was retrieved. We added the following paragraph in the results section (lines 268-271):

“A further analysis was directed to compare the prevalence and severity of placental inflammation in presence or absence of maternal COVID-19 pneumonia. No statistically significant difference was observed for CA, F, chorionic disc inflammation, IFD, and inflammatory infiltrate (tables S4-S8)”.

In order to explain the high number of placentas with chorioamnionitis and funisitis we made the hypothesis of cytokine involvement as a possible reason. Although immune response and cytokine secretion in SARS-CoV-2 pregnancy are still to be clarified, we hypothesized a possible correlation with cytokines activation and added the following part (lines 358-363):

“However, cytokine expression in pregnant women with SARS-CoV-2 infection is yet to be fully understood. In vitro studies detected different cellular immune reactions to SARS-CoV-2 and cytokine secretion in pregnant women compared to non-pregnant women. Especially IL-8 concentration, which has a chemotactic effect on neutrophils, and it is increased in positive SARS-CoV-2 women [25]”, where a new reference was also included (Gomez-Lopez, N.; Romero, R.; Tao, L.; Gershater, M.; Leng, Y.; Zou, C.; Farias-Jofre, M.; Galaz, J.; Miller, D.; Tarca, A.L.; et al. Distinct Cellular Immune Responses to SARS-CoV-2 in Pregnant Women. J Immunol 2022, 208, 1857-1872, doi: 10.4049/jimmunol.2101123.)

As this remains a possibility, we modified the conclusions as follows (lines 475-478):

“The detected high incidence of signs of inflammation, including CA, F, IFD, and VUE could be attributed to a possible immune response induced by the virus, although the effect of SARS-CoV-2 in the placenta is still unclear and under investigation”.

Reviewer 3 Report

Thanks to the authors for the manuscriot 'Placental characteristics of a large Italian cohort of SARS-CoV- 2 2 positive pregnant women'. This study is very relevant and comprehensive, and makes a significant contribution to understanding changes in the placenta with clinical correlation in maternal SARS-CoV-2 infection.

However, as a small remark, I would like to recommend to emphasize the intervillous fibrin deposits as a sign of maternal circulatory disorders and secondary fetal circulatory disorders more clearly, both in the results and in the discussion.

Author Response

We thank the reviewer for the remark “to emphasize the intervillous fibrin deposits as a sign of maternal circulatory disorders and secondary foetal circulatory disorders more clearly, both in the results and in the discussion.”

We introduced the following paragraph in the discussion section, (lines 382-391):

“Moreover, CHI was associated with intrauterine foetal death (IUFD) and early onset intrauterine growth restriction (IUGR). IFD, when massive, is also known as massive perivillous fibrin deposition (MPFD) or maternal floor infarct (MFI). In general, IFD can significantly obstruct maternal blood flow with adverse foetal consequences such as renal dysplasia, stillbirth, IUGR, preterm birth, and neonatal neurological impairment [27]. However, in our series, there was no significant association with SGA neonates and/or NICU admission. Even Stenton did not find substantial neonatal morbidities [20]. The so called “SARS-CoV-2 placentitis” and how it may affect the foetus is still unclear. Probably the timing of maternal viral infection may be a factor in explaining newborns’ outcome”.

Reviewer 4 Report

This is a very interesting study and the largest seria of placentas analysed. The manuscript provides a considerable amount of information concerning the role of placental inflammation in the outcome of pregnancy. Surprisingly placental inflamation did not show any impact on pregnancy complications such as SGA, preterm birth and NICU admission. Obesity and the non-wild type of the virus were the most aggravating factors. Overall, the study is well designed and conducted. The results are thoroughly presented and statistically analyzed. Review of the existing literature is adequate and conclusions are supported by the results. I have to acknowledge the expertise of the participating pathologists in the perinatal field, with consistent assessment of specimens, otherwise a study of this nature would be impossible.

Author Response

We really thank the reviewer for this comment.